# Long COVID: Molecular Mechanisms and Detection Techniques

**DOI:** 10.3390/ijms25010408

**Published:** 2023-12-28

**Authors:** Adela Constantinescu-Bercu, Andrei Lobiuc, Olga Adriana Căliman-Sturdza, Radu Cristian Oiţă, Monica Iavorschi, Naomi-Eunicia Pavăl, Iuliana Șoldănescu, Mihai Dimian, Mihai Covasa

**Affiliations:** 1Department of Biomedical Sciences, Faculty of Medicine and Biological Sciences, “Ştefan cel Mare” University of Suceava, 720229 Suceava, Romania; adela.constantinescu@usm.ro (A.C.-B.); olga.caliman-sturdza@usm.ro (O.A.C.-S.); monica.iavorschi@usm.ro (M.I.); naomi.paval@usm.ro (N.-E.P.); mcovasa@westernu.edu (M.C.); 2Suceava Emergency Clinical County Hospital, 720224 Suceava, Romania; 3Integrated Center for Research, Development and Innovation for Advanced Materials, Nanotechnologies, Manufacturing and Control Distributed Systems (MANSiD), Ştefan cel Mare University of Suceava, 720229 Suceava, Romania; radu.oita@usm.ro (R.C.O.); iuliana.soldanescu@usm.ro (I.Ș.); dimian@usm.ro (M.D.); 4Department of Computers, Electronics and Automation, Ştefan cel Mare University of Suceava, 720229 Suceava, Romania; 5Department of Basic Medical Sciences, College of Osteopathic Medicine, Western University of Health Sciences, Pomona, CA 91711, USA

**Keywords:** immunity, inflammation, SARS-CoV-2, gene expression, sequencing

## Abstract

Long COVID, also known as post-acute sequelae of SARS-CoV-2 infection (PASC), has emerged as a significant health concern following the COVID-19 pandemic. Molecular mechanisms underlying the occurrence and progression of long COVID include viral persistence, immune dysregulation, endothelial dysfunction, and neurological involvement, and highlight the need for further research to develop targeted therapies for this condition. While a clearer picture of the clinical symptomatology is shaping, many molecular mechanisms are yet to be unraveled, given their complexity and high level of interaction with other metabolic pathways. This review summarizes some of the most important symptoms and associated molecular mechanisms that occur in long COVID, as well as the most relevant molecular techniques that can be used in understanding the viral pathogen, its affinity towards the host, and the possible outcomes of host-pathogen interaction.

## 1. Introduction

The ongoing COVID-19 pandemic caused by the severe acute respiratory syndrome coronavirus 2 (SARS-CoV-2) resulted in millions of infections worldwide. While the majority of COVID-19 patients experience mild to moderate symptoms or recover within a few weeks, a significant proportion develop persistent symptoms that can last for months. This condition is known as long COVID or post-acute sequelae of SARS-CoV-2 infection (PASC). A critical aspect of long COVID is the persistence of SARS-CoV-2 in various tissues with several studies detecting viral RNA in multiple organs, including the lungs, heart, and brain, even after the clearance of the virus from the respiratory tract. This viral persistence may be due to a variety of mechanisms, such as the ability of SARS-CoV-2 to establish reservoirs in immune-privileged sites or the capacity of the virus to evade the host immune response [1].

As a result of viral persistence, long COVID is often associated with immune dysregulation, characterized by chronic inflammation, and altered immune responses. Data suggests that a crucial role in the pathogenesis of long COVID is played by dysregulated cytokine signaling, such as overproduction of the triad made of interleukin-1 (IL-1), interleukin-6 (IL-6), and tumor necrosis factor-alpha (TNF-α). A role was proposed for type I IFNs early in the disease and the severe clinical outcomes that occur in participants. Type II interferon signaling and canonical NF-κB signaling (particularly associated with TNF) appear to be the most differentially enriched signaling pathways, and a persistent inflammatory protein signature in the blood can be explained by the persistent presence of viral products [2]. Moreover, interferon (IFN)-γ was selected as a Long Covid biomarker, among 239 candidate markers [3]. Chronic inflammation can lead to tissue damage and contribute to persistent symptoms, including fatigue and muscle pain [4]. Among investigated directions, evidence points to endothelial dysfunction as another key molecular mechanism in long COVID, as endothelial cells play a vital role in regulating blood flow and immune responses. SARS-CoV-2 can directly infect endothelial cells, leading to endothelial damage and dysfunction. This dysfunction can result in microvascular thrombus formation, increased vascular permeability, and impaired blood flow, potentially contributing to a wide range of symptoms, including brain fog and cardiac complications [5] (Figure 1). Furthermore, long COVID often involves neurological symptoms, such as cognitive impairments, headaches, and anosmia. Recent studies suggest that SARS-CoV-2 can directly affect the central nervous system (CNS) by entering the brain through the olfactory nerve or by inducing neuroinflammation. Another entry point of the virus in the brain is systemic circulation since SARS-CoV-2 was detected in human brain tissue and shown to infect vascular endothelial cells and cross into the brain transcellularly through the permeable blood-brain barrier endothelium. This allows for both virus entrance and inflammatory cytokine penetration [6]. While ACE2 is the cellular entry point, it is expressed in choroid plexus cells within the brain ventricles and pericytes throughout the cerebral microvasculature, with low to no expression throughout the rest of the brain. Another factor shown to regulate neural entry is neuropilin 1 (NRP 1) [7]. Additionally, autoimmune responses triggered by the virus may contribute to neurological symptoms [8]. For instance, the autoimmune antibody reaction appears to be a product of the pronounced immune and inflammatory reaction and the detected autoantibodies can belong to antibodies against extracellular, cell surface and membrane, or intracellular targets, including immunoglobulin G (IgG) and immunoglobulin A (IgA) antibodies against cytokines. Such immune effectors can affect cytokine function and endothelial integrity, leading to fatigue and myelitis, and may induce an abnormal renin–angiotensin response, or cause malignant hypertension-related ischemia [9]. Further, autoantibodies appear to be at much higher levels in long COVID patients, as compared to non-COVID or COVID patients and correlate with the intensity of cognitive and physical impairments [10]. These examples indicate that long COVID remains a complex and poorly understood condition with a wide range of clinical manifestations. Molecular mechanisms such as viral persistence, immune dysregulation, endothelial dysfunction, and neurological involvement likely play significant roles in its occurrence and progression. These mechanisms are interconnected, making it challenging to pinpoint a single cause or treatment for long COVID. Thus, further research is needed to elucidate the complexity of long COVID and develop targeted therapies to alleviate its symptoms and improve the quality of life for affected individuals. A multidisciplinary approach, aiming at uncovering and explaining the mechanisms involved, will be essential in understanding long COVID and in developing effective interventions. As such, the scope of the present paper is to offer insight into some of the main molecular processes involved in long COVID occurrence and associated analysis techniques. This is by no means an exhaustive list, as there are daily reports on new regulatory pathways, as well as genetic mechanisms that may be involved in this disease.

## 2. Altered Gene Expression Profiles in Long COVID 

While long COVID clinical symptoms are somewhat more or less clear, the identification of dysregulated genes as part of the molecular mechanisms involved in its pathogenesis offers promising avenues for targeted therapies and personalized treatment strategies. Recent research has begun to uncover changes in gene expression that may contribute to the pathophysiology of this condition. For example, work into the immunobiology of long COVID reveals four main hypotheses behind its pathophysiology: persistent viral antigens leading to chronic inflammation; initiation of an autoimmune response; dysbiosis of microbiome or virome; and impaired tissue repair [11]. Nevertheless, genetic differences in the host may imply variability in the response, for example, the HLA subtypes. A significant difference in the allele frequency of HLA-DRB1*04:01 in severe patients compared to asymptomatic individuals was found, whereas a significantly lower frequency of the haplotype DQA1*01:01-DQB1*05:01-DRB1*01:01 was present in the asymptomatic group compared to the background population. Such genetic traits are useful to know for optimizing treatment as, for instance, HLA-DRB1*04:01 alleles are found in greater frequencies in the North Western European population [12].

### 2.1. Long COVID and Altered Immune Response 

There are two proposed mechanisms behind an altered immune response in long COVID: an ongoing immune response against persistent viral antigens, such as the Spike (S1) protein [13], and immune cell reprogramming [14]. Moreover, studies found that long-term infection with SARS-CoV-2 influences the immune response by increasing the number of monocytes, neutrophils, and CD4+ T-cells while decreasing the amount of CD8+ T-cells and total lymphocytes. The overexpression of pro-inflammatory cytokines, such as interleukin-6 (IL-6), tumor necrosis factor-alpha (TNF-α), interleukin 1b (IL-1b), and interferon-induced protein 44 (IFI44), is commonly observed in long COVID patients, suggesting chronic inflammation and immune activation as key drivers of persistent symptoms [11,15,16,17]. Indeed, it was shown that infection with SARS-CoV-2 is associated with a pro-inflammatory transcriptional profile, though the mechanisms behind these transcriptional changes have not been fully elucidated. COVID-19 infection resulted in the upregulation of the NF-κB inhibitor genes (NFKBIA and NFKBIZ), the AP-1 transcription factor complex genes (FOS, JUN, FOSB, and JUNB), and NF-κB target genes (RELB, NR4A2, DUSP1, and CD69), suggesting an increase in NF-κB-mediated inflammatory response [18]. This is in contrast to an influenza infection which induces an upregulation in type I interferon-mediated inflammatory reaction [19]. Convalescent COVID-19 patients show a pro-inflammatory transcriptional change of monocyte-derived macrophages, as well as an increased reactivity to stimuli, such as lipopolysaccharide (LPS). Among the modified genes, CCL2, CCL7, and CCL8 were increased in post-COVID-19 monocytes, similar to levels in monocytes from patients with severe acute forms of COVID-19. This suggests long-term transcriptional reprogramming, supporting the assertion that long COVID patients present an altered immune response. Aside from these gene modifications which lead to increased chemokine production promoting neutrophil recruitment, other genes have been found to be upregulated post-COVID. These include endothelin-1 (EDN-1), which promotes macrophage activation; FCGBP, which is involved in the anti-viral response; and CYB5R2, which is important in fatty acid metabolism. On the other side of the spectrum, anti-inflammatory mediators were down-regulated in post-COVID such as macrophages, including SEMA7A, nerve growth factor receptor (NGFR), and X inactive specific transcript (XIST) [20]. In addition to increased reactivity to LPS, monocyte-derived macrophages isolated post-COVID also presented an exaggerated response to the S-protein, an effect that persisted for several months after the acute infection. This was similar to the reaction produced by interferon, thus, suggesting the ability of S-protein to induce an anti-viral response post-COVID. Among the upregulated genes were the interferon-stimulated genes (ISGs), IFI27, IFITIM1/3, APOBEC3A, ISG20, OAS1/3, and MX1/2 [13]. Interestingly, interferon-stimulated genes were also downregulated in critical COVID-19 patients [16].

Another immune mechanism involved is a dysfunctional T-cell response. An in-depth analysis involving multi-omic assays and serological tests on individuals with distinct LC and non-LC trajectories, conducted 8 months following infection, highlighted significant differences in T-cell subset distribution. This study revealed an increased frequency of CD4+ T-cells prepared for migration to inflamed tissues and signs of exhaustion in SARS-CoV-2-specific CD8+ T-cells in LC patients. Furthermore, LC individuals displayed elevated SARS-CoV-2 antibody levels and a misalignment between SARS-CoV-2-specific T- and B-cell responses, indicating a breakdown in the crosstalk between the humoral and cellular arms of adaptive immunity. This discoordination might contribute to the immune dysregulation, inflammation, and clinical symptoms observed in long COVID [21]. Additionally, immune dysregulation in long COVID, particularly in individuals who had mild acute COVID-19, includes alterations in T-cells, such as T-cell exhaustion, reduced CD4+ and CD8+ effector memory cell numbers, and elevated PD1 expression on central memory cells, persisting for over a year. Other notable findings include highly activated innate immune cells, a decrease in naive T- and B-cells, and elevated expression of type I and type III interferons. A comprehensive study contrasting patients with long COVID against uninfected individuals and those infected without developing long COVID revealed an increase in non-classical monocytes, activated B cells, double-negative B cells, and IL-4- and IL-6-secreting CD4+ T-cells, alongside a decrease in conventional dendritic cells and exhausted T-cells, and lower cortisol levels in long COVID patients. These immunological perturbations are evident up to 14 months post-infection [22]. In a study that assessed immune activation in COVID-19 patients at 3–12 months post-hospital admission, patients with severe disease exhibited persistent activation of CD4+ and CD8+ T-cells. This activation was evidenced by the expression of HLA-DR, CD38, Ki67, and granzyme B, and elevated plasma levels of interleukins such as IL-4, IL-7, and IL-17, and tumor necrosis factor-alpha (TNF-α). It was observed that plasma from these severe patients could induce T-cells from healthy donors to upregulate IL-15Rα, suggesting an increased T-cell responsiveness to IL-15-driven bystander activation. Interestingly, the study found that the number of long COVID symptoms reported by patients with severe disease did not correlate with cellular immune activation or pro-inflammatory cytokines after adjusting for variables such as age, sex, and disease severity, suggesting that long COVID and persistent immune activation may be independently correlated with severe disease [23].

### 2.2. Long COVID and Impaired Tissue Repair 

Long COVID patients have been experiencing persistent lung and cardiac damage. This has been associated with altered gene expression implicated in tissue repair and regeneration, such as fibroblast growth factor 2 (FGF2) and transforming growth factor-beta (TGF-β), indicating potential disruptions in tissue healing processes [24,25]. SARS-CoV-2 enters host cells via the angiotensin-converting enzyme 2 (ACE2) receptor and the transmembrane serine protease 2 (TMPRSS2). Studies have shown that altered expression of ACE2 and TMPRSS2 may contribute to long COVID pathophysiology. Persistent ACE2 downregulation in certain tissues may affect viral clearance, while TMPRSS2 dysregulation could influence viral entry and host immune responses. Not surprisingly, the severity of COVID-19 has been associated with polymorphisms in ACE2, which promote Spike protein interactions [26], and in TMPRSS2 (transmembrane serine protease 2) [27]. It is, thus, speculated, that these polymorphisms might play a role in the pathogenesis of long COVID as well [9]. In addition, while myocarditis is frequently overlooked during the COVID-19 diagnosis, it occurs due to direct cellular damage and T-cell-mediated cytotoxicity resulting in elevated cardiac biomarkers and is associated with decreased prognosis [28]. It appears that a gene expression alteration occurs in long COVID-associated myocarditis, as four genes are upregulated: Vascular Endothelial Growth Factor A (VEGFA), Forkhead Box O1 (FOXO1), C-X-C Chemokine Receptor 4 (CXCR4), Mothers Against Decapentaplegic Homolog 4 (SMAD4), and two are downregulated: Kirsten Rat Sarcoma Viral Oncogene Homolog (KRAS) and Thioredoxin (TXN) [29].

### 2.3. Long COVID and Neurological Impairments

Neurological clinical manifestations are the hallmark of long COVID symptoms. These include, but are not limited to, cognitive impairments, headaches, dysautonomia, anosmia, hypogeusia, peripheral neuropathy, fatigue, and brain fog [9]. Patients presenting cognitive symptoms after COVID-19 infection showed increased levels of the CCL11 cytokine, which is associated with the recruitment of eosinophils into sites of inflammation and participates in innate immunity. This chemokine exerts physiological and pathological functions in the central nervous system and has been involved in neuroinflammation and impaired hippocampal neurogenesis, while also being linked to impaired myelinating oligodendrocytes. These effects were present in patients suffering from long COVID, despite presenting only mild respiratory symptoms during the acute phase of infection [30]. Responsible for neurological symptoms such as brain fog include cardiovascular factors, with a recent study identifying the presence of a hypercoagulable state or microclots in long COVID patients as one of the culprits for this symptom [31]. 

Among the biomarkers associated with neuronal degradation and damage in long COVID are the neurofilament light chain (NFL) and glial fibrillary acidic protein (GFAP). These are skeletal proteins that are linked to nerve injuries and associated with the severity of long COVID headaches [32,33]. For example, when comparing long COVID patients with or without neurological symptoms, plasma GFAP levels were found to be significantly higher in patients exhibiting central nervous system clinical manifestations, and it was shown to correlate with several immune activation markers during early recovery [33]. Importantly, severe COVID-19 infection has been linked to a subsequent upregulation of the gene profile for Alzheimer’s disease risk. Using a mouse model, Green et al. showed that mice infected with SARS-CoV-2 presented an increase in the expression of the interferon-inducible gene (Ifi204), tau aggregator FKBP51, and complement genes C4 and C5AR1. The majority of the genes involved in Alzheimer’s disease increased following SARS-CoV-2 infection and are also involved in neuroinflammation (CXCL8, EGFR, IL-17, IL-18, IL-6R, and LGALS3), while others regulate neuronal apoptosis (KLF4), tau phosphorylation (FKBP5), or glial cell activation (GFAP, EGFR) [34].

## 3. Mechanisms Leading to Altered Gene Expression in Long COVID

The complete mechanisms involved in long COVID-altered gene expression are far from being understood and their diversity is just beginning to unravel, with miRNAs, transcriptional factors, and lncRNAs, in particular, playing pivotal roles in shaping the clinical outcomes of long COVID. 

### 3.1. Dysregulated miRNA Profiles

Emerging evidence suggests that miRNA dysregulation plays a critical role in long COVID. miRNAs are small non-coding RNA molecules that post-transcriptionally regulate gene expression. Studies have revealed altered miRNA profiles in long COVID patients, with specific miRNAs implicated in immune responses, inflammation, and tissue repair [35]. Several miRNAs, such as miR-146a and miR-155, that have a role in ACE2 expression regulation, have been found to be dysregulated in long COVID patients. These miRNAs are known to modulate immune responses and are key regulators of inflammation-related mediators. Dysregulated miRNA-mediated immune modulation may contribute to the prolonged immune activation observed in long COVID [36,37]. Likewise, molecules involved in tissue repair and fibrosis and miRNAs such as miR-21 and miR-29 have also been associated with persistent organ damage and impaired tissue healing in long COVID [38,39]. For example, the involvement of ACE2-related miRNAs in COVID-19-associated pathologies was explored [40], and, given that certain miRNAs are involved in ACE2 expression regulation in the kidneys (miR-18 and miR-125b), lungs (miR-4262), and heart (miR146a), understanding the miRNAs regulating ACE2 expression can shed light on the organ complications observed in some long COVID patients.

### 3.2. Transcriptional Factors

Epigenetic mechanisms, including DNA methylation and histone modifications, influence gene expression. Recent studies have indicated epigenetic changes in long COVID, with altered DNA methylation patterns observed in genes related to inflammation and immune responses [41,42]. Transcription factors are key regulators of gene expression, and dysregulated transcription factors, such as NF-κB and STAT3, have been identified in long COVID patients. These factors play pivotal roles in cytokine production and immune activation, contributing to the persistent inflammatory state [43,44]. A complex interplay exists between miRNAs and transcriptional factors in long COVID including dysregulated miRNAs targeting transcription factors, subsequently modulating their expression and activity. This intricate crosstalk can further amplify or attenuate immune responses and inflammation, contributing to the pathophysiology of long COVID. The role of numerous families of transcription factors (TFs) in COVID-19 has been already explored. Nevertheless, some changes in human physiology caused by COVID-19 infection or vaccination were only temporary. For example, vaccines against COVID-19 were associated with temporary disruptions of menstrual cycles [45]. Similar changes in menstrual patterns were reported in women infected with COVID-19 [46], which is consistent with other retroviruses [47]. A meta-analysis of gene expression profiles revealed the essential role of several transcription factors on menstrual cycle regularity (IRF1, STAT1, RelA, STAT2, and IRF3) by modulating the prolactin signaling pathways [48]. Specific gene variants in the leucine zipper transcription factor like-1 gene (LZTFL1) were associated with a higher risk of long COVID [49], which could significantly impact the epithelial cells through the activation of the endothelial-mesenchymal transition (EMT), a possible viral response pathway [50].

A major unknown is the organ and tissue distribution of the TFs associated with COVID-19. An interesting discovery is the identification of a list of 19 TFs regulating the expression of a network of 31 gene products directly interacting with the Spike protein of COVID-19, especially in the blood, heart, lung, nasopharynx, and respiratory tract [51]. Several TFs in the list belonged to the Krüppel-like factor family (KLF), which raises the prospect of redundant or overlapping regulation. Notably, immunofibrosis in human lung fibroblasts was accompanied by the downregulation of KLF2 through the activation of JAK-1/2 and IL-6 pathways [52], while its overexpression in endothelium reduced the degree of inflammation [53]. KLF5 was identified as one of the TFs shared by both men and women with COVID-19 [54], a known anti-viral regulator with a lower expression in the lung epithelium from patients with a more severe form of the disease [55]. Immunothrombosis is a major COVID-19 complication [56], and the infection changes the expression levels of key TFs, such as YBX1 and UBTF [57]. In another study, the increase in the platelets GR levels exerted significant effects on the activation of platelets through the non-genomic regulation of post-transcriptional gene expression [58]. A higher platelet activity can lead to severe forms of COVID-19 through interactions with other platelets or leukocytes by aggregation, spreading, and adhesion, amplifying the dysfunction of the endothelium [59]. A recent GWAS study identified FOXP4 as a locus associated with long COVID [60], a gene involved in ciliogenesis and mucus production in the epithelium [61], [62] and the effector cytokine production by T-cells during specific antigen recall responses. As such, TFs can perform different roles in several tissues or cell lineages during COVID-19 infection with synergistic or contradictory effects on the immune responses, simultaneously or sequentially. 

### 3.3. Long Noncoding RNAs

Long non-coding RNAs (lncRNAs) are a class of RNA molecules that do not code for proteins but play crucial roles in regulating gene expression. The immune response to SARS-CoV-2 is multifaceted, and the dysregulation of certain lncRNAs can potentially influence the severity and duration of the disease. For instance, a study on the peripheral immune response in COVID-19 patients identified significant changes in the transcriptional landscape, including the expression of specific lncRNAs [63]. Furthermore, a comprehensive analysis of bronchial epithelial cells infected with SARS-CoV-2 highlighted the role of interferons and the significant changes in the expression of both protein-coding and lncRNAs [64], suggesting their potential role in modulating the cellular response to SARS-CoV-2. While only a small part of the genome encodes for translated gene products, a significant fraction is transcribed into a large variety of RNA species. Long non-coding RNAs (lncRNAs) are longer than 200 bp and are encoded by intergenic regions, while circular RNAs (circRNAs) are microRNAs with the 5′ and 3′ ends joined together [65]. LncRNAs perform several different or overlapping functions, such as post-transcriptional gene regulation, RNA maturation and transport, chromatin remodeling, and molecular decoys. An example is the Metastasis Associated Lung Adenocarcinoma Transcript 1 (MALAT1), a conserved lncRNA localized in the nucleus that was initially identified in highly metastatic tumors [66]. For example, MALAT1 was downregulated in proliferating T-cells from severe COVID-19 patients, [67] while in another study, MALAT1 had a significantly lower overexpression level in the PBMCs from the severe group compared to the mild group [68]. Since MALAT1 was shown to have a protective role in an LPS-induced rodent model of acute lung injury (ALI), [69] a lower level of its expression should theoretically correlate with more inflammation.

Nuclear paraspeckle assembly transcript 1 (NEAT1) is another lncRNA involved in the regulation of immune responses through the formation of paraspeckles and molecular sequestration [70]. NEAT1 was overexpressed in both mild and severe forms of COVID-19 [71]. Interestingly, NEAT1 was increased in saliva samples of patients with COVID-19, thus, being a possible biomarker of the disease [72]. It was also shown that NEAT1 promotes the assembly and activation of inflammasomes in macrophages upon its translocation in the cytosol [73]. TNFα and the heterogeneous nuclear ribonucleoprotein L-related immunoregulatory lincRNA (THRIL) are the lncRNAs promoting cell foam formation and inflammation with its ectopic overexpression triggering the expression of pro-inflammatory genes [74]. THRIL expression was higher in patients with COVID-19 during the post-acute phase compared to the controls, but it was nevertheless decreased during the post-acute phase [75]. However, an atomized analysis of individual lncRNAs with a dysregulated expression profile during a particular pathology might miss the complex interplays between them. Therefore, it is imperative to shift from a modular approach to an integrative network analysis. For example, a recent paper identified the existence of separate clusters of lncRNAs with a lower or higher level of expression during COVID-19 and assigned a risk score to each subtype that can predict the severity of the disease [76].

## 4. Implications of Altered Molecular Processes in Long COVID Symptoms

Long COVID is a syndrome that usually remains undiagnosed because of a multitude of symptoms that can exceed 200 [22]. The most common associations are neurological, with chronic fatigue long after the virus infection has occurred, as well as headaches, anxiety, and insomnia. Other high-risk conditions such as depression, lung problems (breathing difficulties and intercostal pain), and heart dysfunctions (arrhythmias and deep vein thrombosis) are also characteristics of long COVID. Pulmonary lesions may also persist long term after COVID-19. This may be associated with an overload of cytotoxic functions, including γδT (Gamma delta T-cells) and NK (Natural killer) cells, and an increase in lymphocytes (CD4+ and CD8+). In addition, changes in hemoglobin levels due to lung dysfunction were observed as well as an increase in biomarkers such as C-reactive protein or TNF-α (tumor necrosis factor) and IFNm (interferon). Elevated levels of NK or S-sulfocitein were associated with symptoms such as cough and MDSC (Myeloid-Derived Suppressor Cells) with sputum [77,78]. Several molecular changes have been associated with neurological symptoms. For example, higher levels of SARS-CoV-2 protein-containing exosomes were detected in NDEVs (neuron-derived extracellular vesicles) and ADEVs (astrocyte-derived extracellular vesicles) [79]. Moreover, the presence of protein biomarkers in high amounts, such as GFAP (glial fibrillary acidic protein) and Agrin (AGRN), was detected in neurological long COVID [33]. High levels of inflammatory cytokines were also correlated with neurological dysfunction. Neurological manifestations are the most prevalent in long COVID, representing up to 70% of all symptoms [80,81].

Gastrointestinal (GI) clinical symptoms of long COVID have been associated with an enrichment of cytotoxic CD8+/CD4+, TCR (T-cell receptor) clonotypes. These changes were identified approximately 2.5 months after the onset of symptoms (fluctuations in bowel movements, stomach burning, and nausea). The intestinal microbiota has an important role in the body’s immune response and gut microbiota dysbiosis has been associated with long COVID. This leads to significant reductions in beneficial bacteria, bacterial diversities, low abundance of short-chain fatty acids-producing symbionts, and an increase in the number of pathogens (depleted symbionts). These effects persisted even one year after discharge, virus clearance, and the resolution of respiratory symptoms indicating that gut microbiota may play an important role in long COVID [82].

SARS-CoV-2 entrance in the epithelium is favored when the integrity of tight junctions (TJ), which are formed between neighboring cells and regulate the passage of ions and other small solutes, is altered. During the analysis of TJs in lung epithelial cells, the virus was found to affect the integrity of lung epithelial cells [83]. Additionally, the blood-brain barrier can be disturbed by SARS-CoV-2 and lead to derangements of tight junction and adherens junction proteins [84]. One proposed mechanism is that TJs are associated with complexes that mediate cell adhesion and contain the protein occludin (OCLN), whose potential role could be as an internalization factor in SARS-CoV-2 entry and cell-to-cell transmission [85].

In relation to the urinary system and long COVID, a non-specific accumulation of pf ACE2 in the kidneys has been observed, which is associated with renal failure and may, in some cases, be a consequence of thromboembolism. These symptoms are less common in patients with long COVID, thus, differentiating the specific symptoms of long COVID from other causes can be challenging [86]. Since comorbidities such as diabetes, cardiovascular disease, and hypertension, lead to persistent complications of long COVID it is difficult to ascertain whether long COVID is the sole trigger for these symptoms.

### 4.1. Long COVID and Vascular Dysfunctions

The effects of SARS-CoV-2 infection on the blood coagulation system lead to COVID-19-associated coagulopathies, which appear as a prothrombotic state in severely affected patients [87] (Figure 2). Platelets have an increased tendency to form aggregates and show greater responses to stimuli such as ADP, thrombin receptor activator peptide 6, or thrombin [88]. Similar to acute COVID-19, platelet hyperactivity was confirmed by an increase in the expression of P-selectin on their surface [89,90]. Platelet activation can result from various extrinsic mechanisms such as inflammation, endothelial dysfunction, and possibly direct invasion by SARS-CoV-2 [91]. The platelet transcriptome is also altered in COVID-19, with upregulation of pathways such as MAPK signaling, which has implications for platelet activation and function [92]. Given their involvement in both thrombotic events and immune modulation, platelets represent a potential target for treating thrombotic complications of COVID-19. Indeed, antiplatelet therapies have shown promise in reducing mortality and the severity of thrombo-inflammatory complications in COVID-19 patients [93,94].

A thrombo-inflammatory status has also been proposed in long COVID [95]. In part, this may be due to an increase in platelet-neutrophil interactions, which is known to occur in acute COVID-19. The main characterized interactions occur via P-selectin-PSGL-1, CD40L-CD40, GPIb-Mac-1, and GPIIb-IIIa-SLC44A2 [96]. P selectin is the main platelet receptor for aggregation in COVID-19 [97]. Indirectly, both neutrophils and platelets form macrovesicles through membrane budding [98] and secrete inflammatory mediators such as cytokines and IL-6 [99,100]. COVID-19-associated coagulopathy is a major and complex syndrome, recognized due to the pattern that includes thrombin generation, thrombocytosis or thrombocytopenia, decreased fibrinolysis, and high levels of D-dimers. A pulmonary embolism is the main consequence of COVID-19-associated coagulopathy, but thrombi can appear even in minor manifestations of this [101,102]. Fan et al. reported that the level of D-dimer remains high one year after the acute phase of the disease, with an increased thrombin generation, being more frequent in older patients with severe disease [103]. The increased thrombin generation combined with inhibition of fibrinolysis derives from a continuous process of fibrinolysis of thrombi formed during the acute phase and from an ongoing thrombi formation triggered by endothelial dysfunction and thrombin generation. The hemostatic system reacts to the residual effect of COVID-19. The lungs are the primary location of fibrinolysis and the main source of D-dimer, which suggests that hyperfibrinolysis can occur in the pulmonary extra- and intravascular compartments while a systemic hypofibrinolytic state co-exists [104]. 

Persistent endotheliopathy and increased thrombogenicity are present in long COVID [105,106,107]. Endothelial dysfunction is suggested through elevated levels of thrombomodulin, factor VIII, and Von Willebrand Factor (VWF) [108,109]. VWF is a large plasma glycoprotein responsible for normal hemostasis. Its multimeric size is regulated by ADAMTS13, which has been shown to be moderately reduced in long COVID. The increase in VWF and/or decrease in ADAMTS13, a Willebrand factor-cleaving protease, leads to an increased VWF:ADAMTS13 ratio, which is known to be pro-thrombotic. Indeed, this is associated with impaired exercise capacity [110] and increased thrombogenicity under flow [94]. Additionally, a hypofibrinolytic phenotype and increased thrombin generation are described, leading to a hypercoagulable state [111]. A distinct platelet-neutrophil interaction that mimics an acute infection is seen in long COVID patients [97]. Increased D-dimer and interleukin-6 levels are also observed in long COVID patients and might be indicative of microvascular thrombosis [112,113]. Although they are not common, there is a higher risk, especially in severe cases that go beyond the acute phase of the condition. It is still unknown how vaccinations affect the risk of cardiovascular aftereffects. Gaining an understanding of these elements is essential to deciphering the intricate pathophysiology of PASC and developing practical management approaches. For example, immunothrombotic thrombocytopenia (VITT) has been linked to vaccination with AstraZeneca’s ChAdOx1-nCov-19 vaccine, a recombinant adenovirus vector carrying the SARS-CoV-2 Spike protein, and several cases of abnormal thrombotic events and thrombocytopenia have been reported [114,115]. The thrombo-inflammatory dysregulation seen in long COVID infections may be caused by long-lasting structural alterations, viral persistence, or a dysregulated immunological response [96,116]. Procoagulant states may result from long-lasting structural alterations, most notably endotheliopathy, while vasculopathy preserves platelet hyperreactivity even after virus clearance. The finding of SARS-CoV-2 nucleic acids in different organs months after infection, which correlates with protracted COVID symptoms, suggests viral persistence. 

### 4.2. Long COVID and Inflammation

The SARS-CoV-2 entry into host cells via the ACE2 receptor triggers a cascade of innate and adaptive immune responses, including a cytokine storm characterized by elevated levels of IL-6, TNF-α, and other pro-inflammatory cytokines [117,118]. A dysregulated immune response is typified by altered transcriptional patterns and heightened inflammatory markers, and it can last up to six months. Myeloid-derived suppressor cells and cytotoxic CD4+ T-cells are two characteristics of severe COVID-19 that are still present in PASC [119]. In addition to their ability to facilitate viral coinfections and be associated with persistent symptoms, chronic inflammation, and monocyte activation can contribute to immunopathology. Comprehending these mechanisms is crucial in order to formulate focused therapies for PASC [120]. 

Long COVID has exposed the intricate pathophysiology where chronic inflammation persists well beyond the acute infection phase. The role of neutrophils, particularly Neutrophil Extracellular Traps (NETs), is a burgeoning focus of research due to their association with long-term complications such as pulmonary fibrosis, cardiovascular abnormalities, and neurological dysfunction. Neutrophils, as primary responders, become dysregulated in long COVID, and their NETs continue to exert pathological effects, promoting thrombosis and inflammation [121]. NETosis is triggered by a multitude of stimuli including pathogen-associated molecular patterns (PAMPs), cytokines, and autoantibodies, suggesting that the clearance of NETs or their persistence could underpin the pathogenesis of long COVID. The exact pathways of NET formation, involving cellular proteins such as NE, MPO, and PAD4, reflect a highly complex and nuanced process, underpinning both beneficial host defenses and potential deleterious autoimmune responses [122,123].

In COVID-19, inflammation processes affect multiple organs and systems. SARS-CoV-2 infection primarily affects the pulmonary system but also involves other organs such as the cardiovascular system and their complications, such as pulmonary embolism, thrombosis, myocardial infarction, heart failure, and others [124]. For example, in chronic lung diseases, NETs are known to damage tissue and exacerbate inflammation. This damaging potential is extended in the context of COVID-19, where NETs are implicated in persistent pulmonary complications. The correlation of ongoing NET production with lung disease in long COVID has been supported by evidence of elevated neutrophil markers in these patients [125]. Cardiovascular complications of long COVID include increased rates of thrombosis and atherosclerosis. The involvement of NETs in cardiac inflammation and fibrosis is seen in the context of other viral infections, suggesting a similar pathogenic mechanism in long COVID [126]. Neurological manifestations of long COVID are associated with neuroinflammation and suspected blood-brain barrier disruption, where systemic inflammatory responses may affect CNS-resident cells [127]. Despite the absence of direct viral invasion, the role of systemic inflammation, in which NETs may participate, is evident in neurological symptoms post-COVID-19 [128].

## 5. Molecular Methods of Detection and Quantification with Applications in Long COVID

Molecular methods used in long COVID are employed to analyze the genetic and molecular traits of either the viral pathogen or the human host. The gold standard for diagnosis is the reverse transcription-polymerase chain reaction (RT-PCR), which detects the virus’s RNA. RT-PCR is highly sensitive and specific, involving the conversion of viral RNA into DNA, followed by amplification to detectable levels. This method, however, requires specialized equipment and trained personnel, which can be a limitation in rapid and widespread testing scenarios [129]. 

Rapid antigen tests have become a critical tool in the detection of SARS-CoV-2, especially for quick screening purposes. These tests work by detecting specific proteins, known as antigens, present on the surface of the virus. A sample is collected, typically via a nasal or saliva throat swab, and exposed to a solution that releases these viral proteins. The test strip contains antibodies that bind to these antigens if present. This binding is then visualized, usually as a colored line on the test strip, indicating a positive result. While rapid antigen tests are less sensitive than RT-PCR, they offer the significant advantage of providing results within minutes, making them ideal for point-of-care testing and large-scale screenings in various settings [130].

Emerging technologies, such as CRISPR-based assays and next-generation sequencing (NGS), are also enhancing SARS-CoV-2 detection. CRISPR-based assays, for example, use gene-editing technology to target and identify specific sequences of the SARS-CoV-2 genome, offering a rapid and accurate alternative to traditional methods [131]. NGS, on the other hand, sequences the entire viral genome, providing comprehensive information on the virus, including its variants and mutations. These advanced methods are contributing significantly to the global response to the pandemic by improving detection accuracy and efficiency. Among these methods, PCR (and its derivatives), microarray, and sequencing are the most commonly used as described below.

### 5.1. Digital PCR

Digital PCR (dPCR) is an absolute quantitative method that partitions the PCR reaction into a large number of smaller reactions and collects the intensity of fluorescence signals at the endpoint of each reaction [132,133]. Unlike qPCR, dPCR does not depend on standard curves or relative threshold (CT) values for quantitation [134] and is more tolerant to PCR inhibitors [135]. The high precision and the low limit of detection of dPCR could be used for the detection of small variations of DNA or RNA copy number [136], rare allele mutation [137], and gene fusion [138]. Several studies reported that dPCR showed higher sensitivity and specificity in detecting low viral load of SARS-CoV-2 compared with qPCR [139,140,141]. However, the detectability for the samples with extremely low viral load, such as those obtained from recurrent COVID-19 patients, remains unclear. Furthermore, dPCR has been successfully used to detect SARS-CoV-2 RNA in patients who were RT-PCR negative, suggesting that dPCR could identify cases of long COVID that other methods miss. Alteri et al. [140] demonstrated the utility of dPCR in detecting and quantifying SARS-CoV-2 in nasopharyngeal swabs from suspected COVID-19 patients, even when traditional PCR methods failed to confirm the presence of the virus [141].

### 5.2. Microarray

Transcriptomic insights were obtained from a longitudinal whole-blood transcriptomic analysis using microarray technology to characterize immune response alterations in moderate and severe COVID-19 cases, which are fundamental in understanding long COVID. This provides information on processes involved in the activation of neutrophils, platelets, cytokine signaling, and the coagulation system, and identifies distinct gene expression patterns that may persist post-infection. Additionally, proteomic and cytokine profiling using microarray-based analysis revealed significant dysregulation in various proteins related to hemostasis, metabolism, immune responses, and angiogenesis in post-acute COVID-19 syndrome (PACS) patients. Additionally, cytokine profiling showed upregulation in specific cytokines/chemokines, suggesting persistent inflammatory processes in long COVID [142]. To measure the serological antibody response, an antigen microarray protocol for COVID-19 was developed, allowing the assessment of multiple antigen-antibody interactions simultaneously, and understanding the long-term humoral immune response in COVID-19 survivors, which is a key component of long COVID [143]. Microarrays can also be used in analyzing sphingolipid dysregulation in PACS patients and reveal significant alterations, indicating the role of these lipids in cardiovascular dysfunctions observed in long COVID cases. This finding underscores the importance of lipidomic profiling in understanding the systemic effects of long COVID. In addition, IgG and IgM responses were characterized using SARS-CoV-2 proteome microarray in convalescent patients, providing insights into the systemic view of the specific antibody responses, essential for understanding the immune landscape of long COVID [144].

### 5.3. Sequencing Techniques

Sequencing techniques have become invaluable in the study of gene expression alterations in individuals with long COVID, or post-acute sequelae of SARS-CoV-2 infection. These methodologies offer insights into the molecular dynamics of the disease by analyzing the RNA and DNA extracted from samples. RNA Sequencing (RNA-seq) has been extensively utilized to compare gene expression profiles between COVID-19 patients and healthy donors. It allows researchers to quantify and compare RNA expression levels across the entire genome, thus, identifying genes that are differentially expressed due to the viral infection. RNA-seq has also been crucial in identifying shifts in cell populations and gene expression in severe COVID-19 cases, potentially leading to cytokine storms and immune dysregulation.

Single-cell RNA Sequencing (scRNA-seq) provides a high-resolution view of the transcriptomic landscape by profiling individual cells. This technology has been particularly useful in identifying divergent gene expression patterns associated with different clusters of symptoms in long COVID, thus, revealing multiple potential etiologies and pathophysiological pathways. scRNA-seq enables researchers to dissect the heterogeneity within immune cell populations and the complex immune responses seen in PASC. Finally, metagenome sequencing involves sequencing all genetic material within a sample, including the host and microbial DNA. It has been employed to investigate the gut microbiota in COVID-19 patients, revealing dysbiosis that may contribute to disease progression or symptom persistence.

These sequencing techniques, along with computational analyses, have not only facilitated a deeper understanding of the molecular and genetic underpinnings of long COVID but also provided a framework for exploring potential therapeutic targets and biomarkers for this condition. One of the main applications of sequencing in PASC is microRNAs (miRNAs) detection and quantification. The dysregulation of miRNAs during viral infection makes them potential biomarkers for disease detection and prognosis. Computational studies have highlighted the role of miRNAs as powerful tools against COVID-19, while a validated COVID-19 miRNA signature can aid in the differential diagnosis of the disease [145]. Such signatures can be especially beneficial in cases with ambiguous symptoms or long subclinical phases [146]. Considering that circulating miRNA profiles have been linked to the severity of COVID-19 in hospitalized patients, such profiles can potentially guide clinical decisions and interventions, and also miRNA-based therapies, including mimics and inhibitors, which are being explored as potential antiviral treatments against SARS-CoV-2 [147,148]. Several methods have been developed to detect and quantify miRNAs, each with its advantages and limitations. Generally, the RT-PCR method is renowned for its sensitivity and specificity. Protocols have been developed that allow for the detection and quantification of miRNAs, even in limited sample quantities. However, RT-PCR requires prior knowledge of the miRNAs targeted, and the number of miRNAs that can be analyzed at once is limited. As such, Next-generation sequencing (NGS) has revolutionized the field of genomics, enabling high-throughput and cost-effective sequencing of DNA and RNA molecules. Short-read sequencing is one of the most commonly used NGS techniques for miRNA detection. It involves the sequencing of short DNA fragments, typically ranging from 50 to 300 base pairs. This method is highly efficient and can generate millions of reads in a single run, making it suitable for detecting low-abundance miRNAs. However, one limitation of SRS is its inability to sequence longer RNA molecules, which can be crucial for detecting certain miRNA isoforms. In contrast, long-read sequencing can sequence longer RNA molecules, ranging from 1 to 20 kilobases, which allows for the detection of full-length miRNA precursors. However, it is worth noting that LRS often has a higher error rate compared to SRS, which can impact the accuracy of miRNA quantification.

## 6. Conclusions

Long COVID/PASC is proving to be a multisystemic and multisymptomatic disease, which is continuously exposing a plethora of mechanisms. From flu-like symptoms to cardiovascular or neurological effects, long COVID interacts with a vast array of cellular and molecular pathways and interferes with normal physiological processes, in a manner unlike other, similar, viral pathologies. Understanding such interactions requires a holistic approach, where observed symptoms must be analyzed using a multiple-level framework, from the clinical and immunological assessment of the patient to genetic predisposition and co-existing pathologies. This level of understanding, in turn, begs for a broad range of diagnostic and research tools, with sufficient discriminatory power, that can unequivocally identify the mechanisms, regulatory networks, and causes involved in long COVID manifestations. Currently, molecular techniques are paramount in determining long COVID mechanisms, ranging from classical PCR to high throughput, Next Generation Sequencing. Proper use of such techniques will undoubtedly unravel a clearer picture of the disease, however, keeping in mind the fast viral evolution and the need to adapt the techniques and knowledge to current viral evolutionary trends.

## Figures and Tables

**Figure 1 ijms-25-00408-f001:**
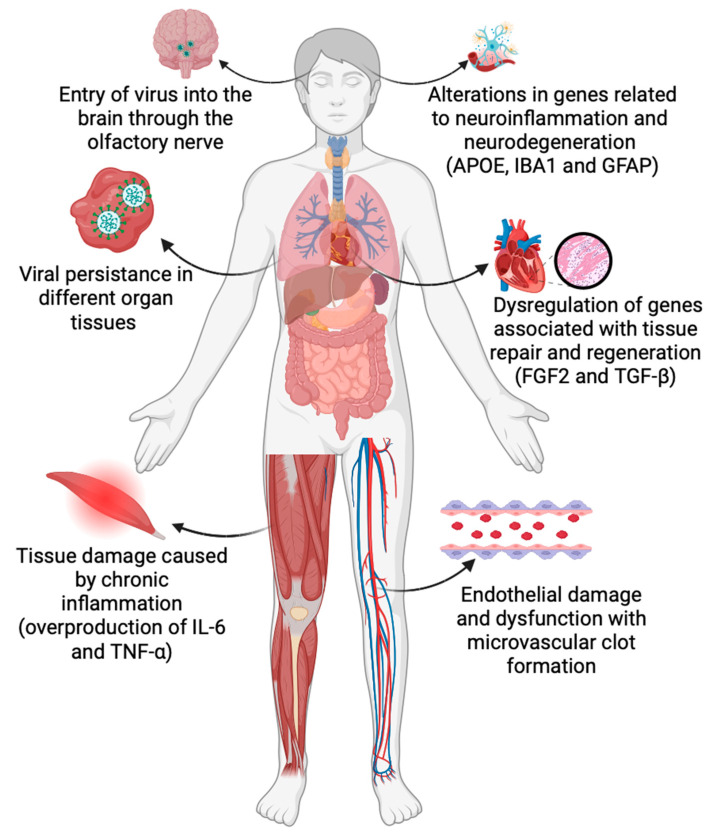
Cellular and molecular mechanisms involved in Long COVID (*abbreviations*: APOE, apolipoprotein E; IBA1, ionized calcium-binding adapter molecule 1; GFAP, glial fibrillary acidic protein; FGF2, Fibroblast Growth Factor 2; TGF-β, Transforming growth factor-β; IL-6, Interleukin 6; and TNF-α, Tumor Necrosis Factor α). (Figure designed using BioRender).

**Figure 2 ijms-25-00408-f002:**
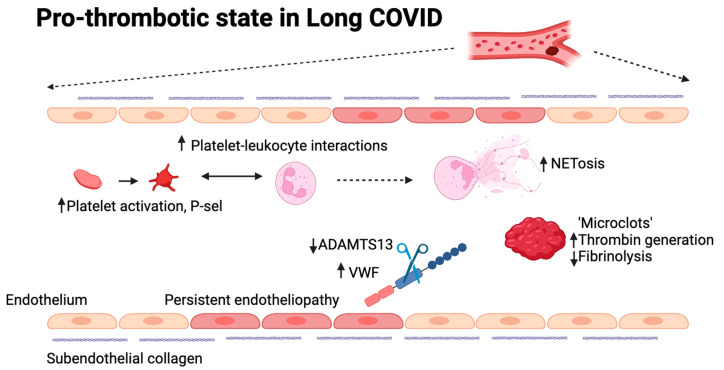
Possible mechanisms involved in the pro-thrombotic state in long COVID. Different mechanisms are currently being explored to understand thrombogenicity in long COVID, many of which have previously been confirmed to play a role in acute COVID-19. These include platelet and neutrophil hyperactivity, increased thrombin generation, decreased fibrinolysis, the presence of microclots, persistent endotheliopathy, and an increase in the VWF:ADAMTS13 ratio. (Figure designed using BioRender).

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
