# Peer review of "Long COVID: Molecular Mechanisms and Detection Techniques"

_ijms, 2023, doi:10.3390/ijms25010408_

Round 1
Reviewer 1 Report
Comments and Suggestions for Authors The paper is well written and does good job in explaining the underlying mechanism involved in Long covid. However, few key points involved in PASC are missing from the manuscript. These issues need to be addressed before considering for the publication. These issues can minor and can be addressed within 1-2 weeks;- Line 42-43, authors have stated the role of Il-6 and TNF-a but have not mentioned about IL-1b, dysregulated IL-1b production is knows to contribute to PASC (PMID: 35732153). Similarly, Interferon-Y response have not been discussed appropriately in the manuscript. Only later in Line 237-238 interferon roles are mentioned in the manuscript. I strongly suggest these two cytokines (Il-1b and IFN) and their roles must be mentioned in the manuscript along with other cytokine wherever possible.
- Line 54-55 how about virus entering the brain from systemic circulation?
- Line 55-56, elaborate autoimmune responses.
- In fig 1, add role of microglia in causing Neuro inflammation ( IBA1) along with GFAP.
- Line 86; authors have not written full forms of abbreviations when introduced in the manuscript for the first time ; spike 1 protein. Later in the manuscript line 210, spike protein is spelled. This inconsistency must be addressed. Another example is line 103 with LPS, 399 with PAMPs.
- Line 136-148, myocarditis is one of the major cardiovascular complications associated with SARS-cov2 infection and authors have failed to mention it in the paragraph. Please add myocarditis in the paragraph and cite appropriately.
- Line 153-154 elaborate the findings explained on the original paper, GFAP was found higher in what sample or cell types ?
- Line 295-304, add role of tight junctions protein in long covid ? Compromised tight junctions protein have been associated with the persistent viral infection. Please address.
- In detection and quantification of long covid, please add few other methods commonly used in detection of the virus, example from saliva.
Author Response
The paper is well written and does good job in explaining the underlying mechanism involved in Long covid. However, few key points involved in PASC are missing from the manuscript. These issues need to be addressed before considering for the publication. These issues can minor and can be addressed within 1-2 weeks.
Thank you for your constructive suggestions which helped improved our paper.
- Line 42-43, authors have stated the role of Il-6 and TNF-a but have not mentioned about IL-1b, dysregulated IL-1b production is knows to contribute to PASC (PMID: 35732153). Similarly, Interferon-Y response have not been discussed appropriately in the manuscript. Only later in Line 237-238 interferon roles are mentioned in the manuscript. I strongly suggest these two cytokines (Il-1b and IFN) and their roles must be mentioned in the manuscript along with other cytokine wherever possible.
The IL-1b and IFN gamma were added. The suggested citation as well as other appropriate references were included
Line 54-55 how about virus entering the brain from systemic circulation?
The information has been added
- Line 55-56, elaborate autoimmune responses.
Autoimmune response details were added in several places
- In fig 1, add role of microglia in causing Neuro inflammation ( IBA1) along with GFAP.
IBA1 was added to Fig. 1
- Line 86; authors have not written full forms of abbreviations when introduced in the manuscript for the first time ; spike 1 protein. Later in the manuscript line 210, spike protein is spelled. This inconsistency must be addressed. Another example is line 103 with LPS, 399 with PAMPs.
Abbreviations were defined at first occurence. Grammatical errors were corrected.
- Line 136-148, myocarditis is one of the major cardiovascular complications associated with SARS-cov2 infection and authors have failed to mention it in the paragraph. Please add myocarditis in the paragraph and cite appropriately.
Myocarditis and appropriate reference were added
- Line 153-154 elaborate the findings explained on the original paper, GFAP was found higher in what sample or cell types ?
This information was expanded
- Line 295-304, add role of tight junctions protein in long covid ? Compromised tight junctions protein have been associated with the persistent viral infection. Please address.
The information on the role of TJs was added.
- In detection and quantification of long covid, please add few other methods commonly used in detection of the virus, example from saliva.
Other methods were added including detection from saliva
Overall, I strongly suggest authors to add a paragraphs explaining role of different HLA subtype and another paragraph explaining dysfunctional T cell response contributing to Long covid.
Both paragraphs were added
For the title, it states "occurrance" which also implies prevalence and incidence of long covid. Since, the manuscript is not epidemiology rather cellular and molecular mechanics based, I suggest rephrasing title to better reflect the content of the manuscript. One suggestion would be to remove the word “occurrence “ and read like
The title was changed, as suggested.
Reviewer 2 Report
Comments and Suggestions for Authors
In the manuscript entitled "Molecular Mechanisms Involved in the Occurrence and Progression of Long COVID and Associated Analysis Techniques." the authors have tried to provide a comprehensive review of Long COVID and associated techniques for detection and analysis.
Overall, in my opinion, the manuscript is well-written; however, it needs minor corrections.
Comments:
1. In my opinion, the title can be improved.
2. There is no mention of Figure 2 in the manuscript anywhere.
3. In the "Sequencing" section, in Line 474-477, "NLP has been used in conjunction with expression analysis to interpret complex data sets derived from sequencing." The author has mentioned the use of NLP techniques, but there is no explanation provided on how these techniques are useful in this context. It would be helpful to provide a few examples and cite appropriate references.
Author Response
Thank you very much for your constructive comments which helped improve our paper.
In my opinion, the title can be improved.
The title was changed as also suggested by Reviewer 1
- There is no mention of Figure 2 in the manuscript anywhere.
Figure 2 reference was added
- In the "Sequencing" section, in Line 474-477, "NLP has been used in conjunction with expression analysis to interpret complex data sets derived from sequencing." The author has mentioned the use of NLP techniques, but there is no explanation provided on how these techniques are useful in this context. It would be helpful to provide a few examples and cite appropriate references.
The paragraph was removed, as it had little relevance.